 eLIFE

# Connexin26 hemichannels with a mutation that causes KID syndrome in humans lack sensitivity to $CO_2$

**Louise Meigh[1], Naveed Hussain[2], Daniel K Mulkey[3], Nicholas Dale[1]\***

[1]School of Life Sciences, University of Warwick, Coventry, United Kingdom; [2]Division of Neonatal Pediatrics, Connecticut Children's Medical Center NICU, University of Connecticut Health Center, Farmington, United States; [3]Department of Physiology and Neurobiology, University of Connecticut, Storrs, United States

**Abstract** Mutations in connexin26 (Cx26) underlie a range of serious human pathologies. Previously we have shown that Cx26 hemichannels are directly opened by $CO_2$ (*Meigh et al., 2013*). However the effects of human disease-causing mutations on the $CO_2$ sensitivity of Cx26 are entirely unknown. Here, we report the first connection between the $CO_2$ sensitivity of Cx26 and human pathology, by demonstrating that Cx26 hemichannels with the mutation A88V, linked to Keratitis-Ichthyosis-Deafness syndrome, are both $CO_2$ insensitive and associated with disordered breathing in humans.

**\*For correspondence:** n.e.dale@warwick.ac.uk

**Competing interests:** The authors declare that no competing interests exist.

Connexin26 (Cx26) is one of 21 connexin genes found in humans (*Cruciani and Mikalsen, 2006*). The canonical function of connexins is to form gap junctions in which two hexameric connexons, or hemichannels, in closely apposed membranes dock together to form an intercellular channel. However connexins can also function as hemichannels, thereby providing large conductance channels, which allow passage of small molecules such as ATP into the extracellular space (*Stout et al., 2004*; *Wang et al., 2013*). We have recently shown that Cx26 hemichannels are directly sensitive to $CO_2$ (*Huckstepp et al., 2010a*; *Meigh et al., 2013*). When $CO_2$ binds to Cx26, it carbamylates K125, forms a salt bridge to R104 and opens the hemichannel (*Meigh et al., 2013*). Cx26 hemichannels are thus a source of $CO_2$-gated ATP release (*Huckstepp et al., 2010a*).

Mutations of Cx26 are the commonest cause of non-syndromic hearing loss (*Cohn and Kelley, 1999*; *Kelley et al., 2000*; *Xu and Nicholson, 2013*). Some of these mutations cause loss of functional protein, while other mutations result in gap junctions and hemichannels with altered properties. However the effect of these mutations on the $CO_2$ sensitivity of Cx26 has never been examined. Some missense mutations of Cx26 cause serious pathologies in humans, such as the very rare ectodermal disorder, Keratitis-Ichthyosis-Deafness (KID) syndrome. KID syndrome involves a combination of deafness, visual impairment, and dermatological abnormalities (*Caceres-Rios et al., 1996*). About 100 cases have been reported in the literature, and of these around 70% are caused by de novo mutations in Cx26, with the remainder being inherited in an autosomal dominant manner or via germ line mosaicism (*Sbidian et al., 2010*). To date there are nine missense mutations that can cause KID syndrome (*Xu and Nicholson, 2013*). The severity of the symptoms of KID syndrome depends on the particular mutation in Cx26 (*Janecke et al., 2005*; *Jonard et al., 2008*).

The mutation, Cx26[A88V], is linked to a very severe form of KID syndrome, which is fatal in infancy (*Haruna et al., 2010*; *Koppelhus et al., 2010*). In one of the original reports linking Cx26[A88V] to KID syndrome, the patient required mechanical ventilation (*Koppelhus et al., 2010*), suggesting a possible effect of the mutation on the neural control of breathing. In KID syndrome caused by a different missense mutation (G45E), which is fatal within the first year of life, there are also reports of breathing problems. One patient required mechanical ventilation immediately after birth (*Janecke et al., 2005*)

and a second died from breathing failure (*Sbidian et al., 2010*). Nevertheless, without detailed recordings of cardiorespiratory activity, it is not possible to know whether these patients experienced inadequate central respiratory drive. For other mutations linked to KID syndrome there are no reports of abnormal breathing in the literature.

The reason why the A88V and G45E mutations should cause such pervasive and severe pathology remains unclear as subunits of $Cx26^{A88V}$ and $Cx26^{G45E}$ form both functional gap junctions and hemichannels (*Gerido et al., 2007*; *Mhaske et al., 2013*). Expression of $Cx26^{A88V}$ in HeLa cells gives rise to enhanced hemichannel-mediated currents (compared to wild type Cx26, $Cx26^{WT}$) at positive transmembrane potentials and in the absence of extracellular $Ca^{2+}$, leading to the suggestion that this mutation represents a gain of function (*Mhaske et al., 2013*). The G45E mutation, also causes enhanced hemichannel activity in the absence of extracellular $Ca^{2+}$, and increased permeability to $Ca^{2+}$ (*Gerido et al., 2007*; *Sanchez et al., 2010*). A gain of function has therefore been suggested as underlying the actions of this mutation too. Although the absence of extracellular $Ca^{2+}$ opens connexin hemichannels, this condition is unlikely to occur in physiological systems. Thus the consequences of the A88V and G45E mutations on physiologically relevant gating of Cx26 remain unclear.

We identified a patient with KID syndrome caused by a heterozygous Cx26 A88V mutation. This patient failed to breathe spontaneously at birth and initially required mechanical ventilation. Later when he started to breathe spontaneously, he continued to demonstrate periods of apnea and bradycardia. A pneumogram performed at a post-menstrual age of 40 weeks showed abnormal persistence of central apnea lasting ≥20 s and accompanied by periods of bradycardia and prolonged oxygen desaturation (*Figure 1*). This respiratory pattern is abnormal for the age of the infant and is suggestive of blunted chemosensory control of breathing. Given the previously described role of Cx26 in mediating the $CO_2$-dependent drive to breathe (*Huckstepp et al., 2010b*; *Wenker et al., 2012*), we considered whether the mutation A88V might alter the $CO_2$-sensitivity of Cx26.

We introduced the A88V mutation into Cx26 and then tested the $CO_2$ sensitivity of $Cx26^{A88V}$ hemichannels expressed in HeLa cells via an established and sensitive dye-loading protocol (*Huckstepp et al., 2010a*; *Meigh et al., 2013*). Under conditions of normal extracellular $Ca^{2+}$, HeLa cells expressing wild type Cx26 hemichannels readily load with carboxyfluorescein when exposed to a moderately hypercapnic saline ($PCO_2$ 55 mmHg) (*Huckstepp et al., 2010a*; *Meigh et al., 2013*). However HeLa cells expressing $Cx26^{A88V}$ showed no such $CO_2$-dependent dye loading even when exposed to higher

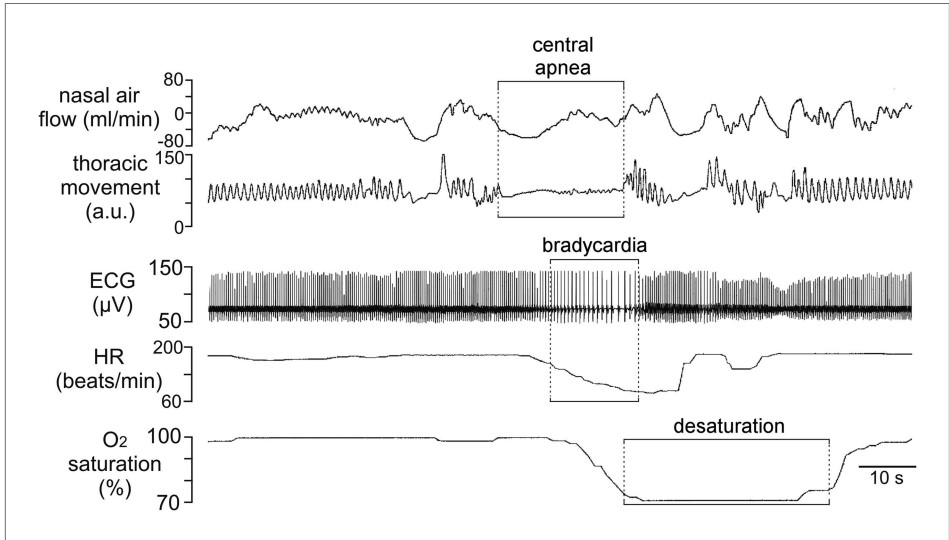

**Figure 1**. Incidence of central sleep apnea in a patient with $Cx26^{A88V}$. Recording of cardiorespiratory activity during sleep from an infant at a post-menstrual age of 40 weeks diagnosed with KID syndrome. Traces of nasal air flow, thoracic movement, electrocardiogram (ECG), heart rate (HR) and arterial $O_2$ saturation show that this patient exhibited a prolonged period during which no effort was made to breathe and this was followed by pronounced bradycardia and arterial $O_2$ desaturation, all of which are characteristic of central sleep apnea. Unfortunately, at 2 months of age this patient died from overwhelming sepsis.

levels of $PCO_2$ (70 mmHg, **Figure 2**). The failure to exhibit $CO_2$-dependent dye loading was not due to a lack of functional hemichannels as the positive control of removing extracellular $Ca^{2+}$, which opens all connexin hemichannels, caused robust dye loading (**Figure 2**). HeLa cells transfected with an empty vector do not show any dye loading in response to a $CO_2$ stimulus or removal of extracellular $Ca^{2+}$ (**Figure 2—figure supplement 1**). Surprisingly therefore, the conservative mutation A88V caused Cx26 hemichannels to lose their sensitivity to $CO_2$. As this mutation is far from the residues involved in $CO_2$ binding (K125 and R104), the mechanism for the loss of $CO_2$ sensitivity is unclear.

As the missense mutations which underlie KID syndrome act in a dominant manner (**Jonard et al., 2008**; **Xu and Nicholson, 2013**), we tested whether the expression of Cx26[A88V] subunits might have a dominant negative action on the $CO_2$ sensitivity of Cx26[WT]. We transfected HeLa cells that stably expressed Cx26[WT] with the Cx26[A88V] subunit and documented their sensitivity to $CO_2$ following transfection. 4 days after transfection with Cx26[A88V], the HeLa cells still exhibited sensitivity to $CO_2$ (**Figure 3A**), but this was reduced compared to the Cx26[WT] HeLa cells that had not been transfected with Cx26[A88V] (**Figure 3B**). 5 and 6 days after transfection, the HeLa cells showed no sensitivity to $CO_2$ (**Figure 3A**). Nevertheless functional hemichannels were still present as the removal of extracellular $Ca^{2+}$ caused dye loading (**Figure 3A**). The loss of $CO_2$ sensitivity was not simply a consequence of days in

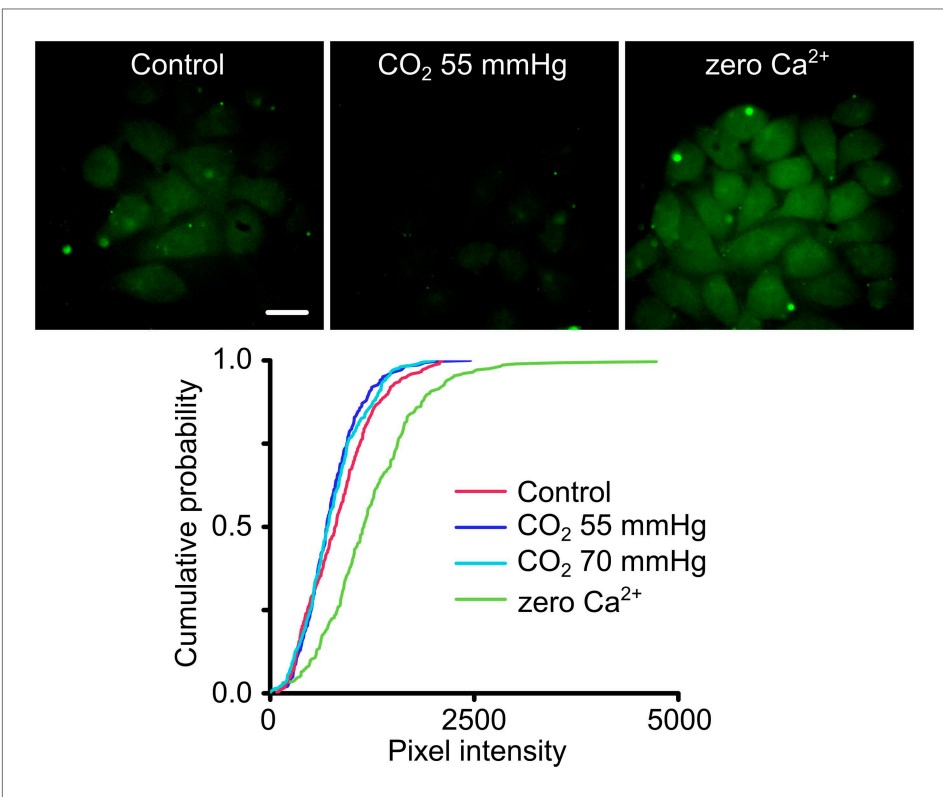

**Figure 2**. Cx26[A88V] hemichannels are no longer sensitive to $CO_2$. (Top) Images of HeLa cells expressing Cx26[A88V] under control, hypercapnic and zero $Ca^{2+}$ conditions. The cells were exposed to 200 µM carboxyfluorescein (CBF) for 5 min under each condition before being washed. Some low background loading of CBF is seen under control conditions. In presence of $CO_2$ no loading is seen. The positive control of removal of extracellular $Ca^{2+}$ causes robust dye loading demonstrating the presence of functional hemichannels. (Bottom) Cumulative probability distributions of pixel intensity of HeLa cells expressing Cx26[A88V] under control, hypercapnia (two levels of $PCO_2$) and zero $Ca^{2+}$. Only the removal of extracellular $Ca^{2+}$ causes dye loading as shown by the rightward shift of the curve to higher pixel intensities (p = 0.004, Mann Whitney U test compared to control). These distributions show all of the measurements made (minimum 40 cells each from five independent repetitions).

The following figure supplement is available for figure 2:

**Figure supplement 1**. HeLa cells transfected with the empty pCAG-GS mCherry vector show no sensitivity to $CO_2$ and do not dye load when exposed to zero $Ca^{2+}$ aCSF.

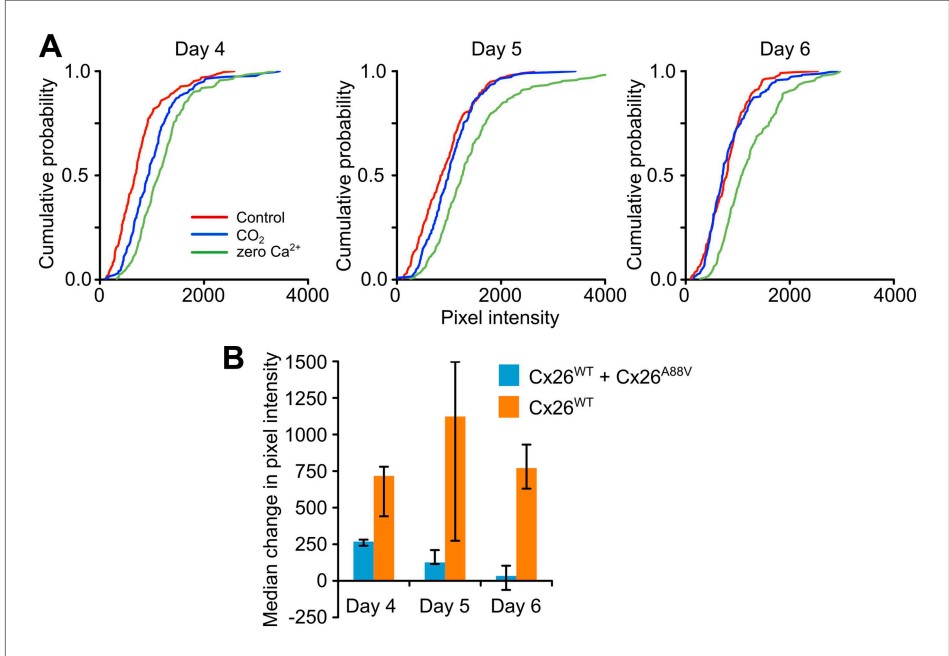

**Figure 3**. $Cx26^{A88V}$ hemichannels act in a dominant negative manner to remove $CO_2$ sensitivity from $Cx26^{WT}$. (**A**) Cumulative probability distributions for $CO_2$-dependent dye loading in HeLa cells that stably express $Cx26^{WT}$, which have been transfected with $Cx26^{A88V}$. 4 days after transfection with $Cx26^{A88V}$ the cells still exhibit significant sensitivity to 55 mmHg $PCO_2$ stimulus (p = 0.048 $CO_2$ compared to control, Mann Whitney U test). 5 and 6 days after transfection the $CO_2$ sensitivity of the HeLa cells was abolished. On all 3 days, the positive control of zero $Ca^{2+}$ caused dye loading, demonstrating the presence of functional hemichannels. The graphs show all of the measurements made from 5 independent repetitions of the experiment. (**B**) Comparison of the sensitivity to $CO_2$ of HeLa cells stably expressing $Cx26^{WT}$ which have been transfected with $Cx26^{A88V}$ ($Cx26^{WT}$ + $Cx26^{A88V}$, n = 5) with those that have not ($Cx26^{WT}$, n = 7). In the absence of transfection, the $Cx26^{WT}$-expresssing HeLa cells retain sensitivity to $CO_2$ on all 3 days. By contrast $Cx26^{A88V}$ causes significantly depressed $CO_2$ sensitivity 4 days after transfection (p = 0.001), and loss of sensitivity on days 5 (p = 0.024) and 6 (p = 0.001). Comparisons of $Cx26^{WT}$ with $Cx26^{WT}$ + $Cx26^{A88V}$ via Mann Whitney U test, and False Discovery Rate procedure for multiple comparisons (***Curran-Everett, 2000***). Error bars upper and lower quartiles.

culture, as $Cx26^{WT}$ HeLa cells that had not been transfected with $Cx26^{A88V}$ retained their sensitivity to $CO_2$ over the whole period examined (***Figure 3B***). We therefore conclude that $Cx26^{A88V}$ subunits are able to act in a dominant negative manner to cause loss of $CO_2$ sensitivity from wild type Cx26 hemichannels.

This is the first instance in which a mutation linked to serious human pathologies has been demonstrated to abolish the $CO_2$ sensitivity of Cx26. This in turn suggests that Cx26-mediated $CO_2$ sensing may be important for human physiology in the range of contexts that are associated with the diverse pathologies linked to this mutation. In the closely related β connexin, connexin30 (Cx30), the mutation A88V connected to Clouston's Syndrome (***Bosen et al., 2014***), may result in constitutively open Cx30 hemichannels (***Essenfelder et al., 2004***). However Cx30 is also opened by $CO_2$ (***Huckstepp et al., 2010a***) and the effect of this mutation on the $CO_2$ sensitivity of Cx30 has not yet been investigated. There are no reports in the literature of disordered breathing in patients with Clouston's syndrome.

Previous studies suggesting that the A88V mutation gave a gain of function in Cx26, examined hemichannel function in the absence of extracellular $Ca^{2+}$ (***Mhaske et al., 2013***). As the $CO_2$ sensitivity of the mutated hemichannel was not specifically examined in this previous study, it is likely that both sets of findings are correct—an enhancement of macroscopic hemichannel currents (***Mhaske et al., 2013***), and a loss of $CO_2$ sensitivity. However under physiological conditions of normal extracellular $Ca^{2+}$ and in the presence of physiological $CO_2/HCO_3^-$ buffering, we suggest that A88V should be considered as a loss-of-function mutation that effectively removes the capacity for $CO_2$-evoked ATP release via Cx26 hemichannels.

Our report is the first to document altered central respiratory drive in a KID syndrome patient. In rodents, $CO_2$-sensitivity of Cx26 contributes to the chemosensory control of breathing (*Huckstepp et al., 2010b*; *Wenker et al., 2012*). Although we do not know if the loss of $CO_2$ sensitivity in Cx26 contributed to the aberrant respiratory drive exhibited by this patient, these results are consistent with this possibility, and represent the first evidence to suggest that Cx26 hemichannels are a requisite component of the drive to breathe in humans. Overall the ability of physiological levels of $PCO_2$ to permit ATP release via Cx26 hemichannels may be important in the epidermis, cochlea and brain. Investigation of whether the absence of this mechanism of ATP release in patients with Cx26[A88V] contributes to the serious pathological abnormalities that they suffer would seem to be warranted.

## Materials and methods

### Case study

The Institutional Review Board of the Connecticut Children's Medical Center considered this under the category of a case report and thus exempt from formal review.

### Mutant connexin production

Puc19 Cx26[A88V] was produced from wild type Cx26 via the Quikchange protocol using the following primers: forward 5′ TGT CCA CGC CGG TCC TCC TGG TAG C 3′ reverse 5′ GCT ACC AGG AGG ACC GGC GTG GAC A 3′. Cx26[A88V] was subcloned into a pCAG-GS mCherry vector for mammalian cell transfection. Successful mutation of Cx26 was confirmed by sequencing which also verified that apart from the desired mutation the sequence was identical to the wild type.

### HeLa cell culture

HeLa cells were cultured by standard methods in DMEM, 10% FCS with addition of 3 mM $CaCl_2$. For experimentation, cells were plated onto coverslips at a density of $5 \times 10^4$ cells per well. Transient transfections were performed using the genejuice protocol.

### Solutions used

#### Control aCSF

124 mM NaCl, 26 mM $NaHCO_3$, 1.25 mM $NaH_2PO_4$, 3 mM KCl, 10 mM D-glucose, 1 mM $MgSO_4$, 1 mM $CaCl_2$.

#### Zero $Ca^{2+}$ aCSF

124 mM NaCl, 26 mM $NaHCO_3$, 1.25 mM $NaH_2PO_4$, 3 mM KCl, 10 mM D-glucose, 1 mM $MgSO_4$, 1 mM $MgCl_2$, 1 mM EGTA.

#### Hypercapnic (55 mmHg $CO_2$) aCSF

100 mM NaCl, 50 mM $NaHCO_3$, 1.25 mM $NaH_2PO_4$, 3 mM KCl, 10 mM D-glucose, 1 mM $MgSO_4$, 1 mM $CaCl_2$.

#### Hypercapnic (70 mmHg $CO_2$) aCSF

70 mM NaCl, 80 mM $NaHCO_3$, 1.25 mM $NaH_2PO_4$, 3 mM KCl, 10 mM D-glucose, 1 mM $MgSO_4$, 1 mM $CaCl_2$.

Hypercapnic aCSF was saturated with sufficient $CO_2$ (the remaining balance being $O_2$) to adjust the final pH (pH 7.5) to that of the control aCSF removing any potential effects of changes in extracellular pH.

All other solutions were saturated with 95% $O_2$/5% $CO_2$.

### Dye loading protocols

Coverslips plated with HeLa cells transiently transfected with Cx26[A88V] were exposed to Hypercapnic aCSF (55 mmHg or 70 mmHg) containing 200 μM CBF for 10 min. This was followed by control aCSF with 200 μM CBF for 5 min and a 30 min wash with control aCSF to ensure that all dye is removed from the outside of the cells.

A control comparison was used to establish any baseline loading occurring in the absence of a stimulus. HeLa cells expressing Cx26[A88V] were exposed to 200 μM CBF in control aCSF for 15 min, followed by 30 min of washing.

A zero $Ca^{2+}$ positive control was also performed to ensure functional connexin hemichannels were being expressed. $Cx26^{A88V}$ expressing HeLa cells were exposed to 200 µM CBF in zero $Ca^{2+}$ aCSF for 10 min. This was followed by control aCSF with 200 µM CBF for 5 min and 30 min of washing with aCSF.

## Imaging and analysis

For each condition cells were imaged by epifluorescence (Scientifica Slice Scope, Cairn Research OptoLED illumination, 60× water Olympus immersion objective, NA 1.0, Hamamatsu ImageEM EMCCD camera, Metafluor software). Using ImageJ, the extent of dye loading was measured by drawing a region of interest (ROI) around individual cells and calculating the mean pixel intensity for the ROI. The mean pixel intensity of the background fluorescence was also measured in a representative ROI, and this value was subtracted from the measures obtained from the cells. All of the images displayed in the figures reflect this procedure in that the mean intensity of the pixels in a representative background ROI has been subtracted from every pixel of the image. The analysis of the $CO_2$ sensitivity of $Cx26^{A88V}$ was performed as five independent repetitions in which at least 40 cells were measured in each condition, and the mean pixel intensities plotted as cumulative probability distributions.

# Additional information

### Funding

| Funder | Grant reference number | Author |
| --- | --- | --- |
| Medical Research Council | G1001259 | Nicholas Dale |

The funder had no role in study design, data collection and interpretation, or the decision to submit the work for publication.

### Author contributions

LM, NH, Conception and design, Acquisition of data, Analysis and interpretation of data, Drafting or revising the article; DKM, ND, Conception and design, Analysis and interpretation of data, Drafting or revising the article

### Ethics

Human subjects: The Institutional Review Board of the Connecticut Children's Medical Center considered this under the category of a case report and thus exempt from formal review.

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
