## [Decision Letter]

Thank you for sending your work entitled “Cx26 hemichannels with the A88V mutation causing Keratitis-Ichthyosis-Deafness syndrome in humans lack CO_2_ sensitivity” for consideration at *eLife*. Your article has been favorably evaluated by Michael Marletta (Senior editor) and 2 reviewers, one of whom, Juan Saez, has agreed to reveal his identity.

The Senior editor has assembled the following comments to help you prepare a revised submission.

Your findings reported here are very interesting; however, you have no direct evidence that there are carbamylation changes in the A88V mutant. Hence, speculation regarding specific mechanism should be removed from the manuscript, since we all agree that your observations may not be the result of changes in carbamylation. We had an active discussion about the value of asking you to include expanded information on the patient and particularly about familial history of the disease in order to link the mutation to the respiratory phenotype. In the end we think it would be informative if you provide information on the prevalence of KID patients with respiratory problems and on their longevity.

---

## [Author Response]

*Your findings reported here are very interesting; however, you have no direct evidence that there are carbamylation changes in the A88V mutant. Hence, speculation regarding specific mechanism should be removed from the manuscript, since we all agree that your observations may not be the result of changes in carbamylation. We had an active discussion about the value of asking you to include expanded information on the patient and particularly about familial history of the disease in order to link the mutation to the respiratory phenotype. In the end we think it would be informative if you provide information on the prevalence of KID patients with respiratory problems and on their longevity*.

We have removed any speculation that the A88V mutation alters carbamylation of Cx26.

We agree that more background information on KID syndrome would be helpful for the reader. This is a complex area as different mutations in Cx26 give forms of the syndrome that differ in their severity. The mutation A88V causes a highly severe form that is fatal in early childhood. In the first report linking A88V to KID syndrome, a requirement for mechanical ventilation was noted; a second report on A88V did not note any respiratory problems. A different mutation, G45E, also caused a severe form of KID syndrome, which was fatal in the first year of life. There are two reports of abnormal breathing in these patients. However no detailed cardiorespiratory recordings were presented in these prior descriptions of KID syndrome, and thus we cannot assess whether the abnormalities in breathing resulted from a loss of central respiratory drive.

For other mutations linked to KID syndrome such as D50N, which generally give a less severe form of the disease, there are no reports of breathing abnormalities in the literature. However we would be cautious in concluding from this that such abnormalities do not exist. It is possible that they were not sufficiently severe to be a clinical problem or were not actively investigated.

To address this issue in the paper, we have summarized the reported incidence of respiratory problems in A88V and G45E patients, and have simply stated that there are no reports of any such problems in KID syndrome patients with other Cx26 mutations.

An interesting feature of the missense mutations that underlie KID syndrome is that they are dominant. The Cx26^A88V^ subunit should therefore act in a dominant negative fashion to remove CO_2_ sensitivity from cells expressing wild type Cx26 hemichannels. We have now had an opportunity to test this idea, and now present this evidence in a new Figure 3. We believe that this additional evidence strengthens the potential link between the loss of CO_2_ sensitivity in Cx26^A88V^ and the pathologies evident in these patients.